# Reliability and Validity of IMU-Based Foot Progression Angle Measurement under Different Gait Retraining Strategies

Francine C. A. Urbanus [1], Jane Grayson [2] , Jaap Harlaar [1,3] and Milena Simic [2,*]

1 Department of BioMechanical Engineering, Delft University of Technology, 2628 CD Delft, The Netherlands; francineurbanus@hotmail.com (F.C.A.U.); j.harlaar@tudelft.nl (J.H.)
2 Discipline of Physiotherapy, Faculty of Medicine and Health, The University of Sydney, Sydney 2006, Australia; jane.grayson@sydney.edu.au
3 Department of Orthopedics, Erasmus Medical Center, 3015 GD Rotterdam, The Netherlands
* Correspondence: milena.simic@sydney.edu.au

**Abstract:** Load modifying gait retraining strategies, such as changing the foot progression angle (FPA) to toe-in and toe-out gait, are used for people with medial knee osteoarthritis. The FPA can be measured using a pressure sensitive walkway (PSW), but inertial measurement units (IMUs) are considered more suitable for clinical use. This study evaluated the reliability and validity of an IMU system, to measure FPA under different gait retraining strategies. Twenty healthy participants walked a 10-m-long path using different gait strategies (natural (2), toe-out gait (1), toe-in gait (1)) during four 90-s trials. FPA was measured simultaneously with IMUs and a PSW, the latter considered the reference standard. There was good and excellent reliability for the IMUs and PSW FPA measurements, respectively (ICC: IMU, 0.89; PSW, 0.97). Minimal detectable change (MDC) was 4.5° for the IMUs and 2.7° for the PSW. Repeated measures ANOVA indicated a significant effect of gait type on FPA ($p < 0.001$), but not the measurement instrument ($p = 0.875$). Bland–Altman plots demonstrated the good agreement of both systems for the baseline condition, though the IMUs seemed to consistently overestimate the FPA value compared to the PSW. In conclusion, IMUs are a reliable and valid measurement system for measuring FPA under different gait retraining strategies. The differences between the systems are significant for all gait strategies, so the systems should not be used interchangeably.

**Keywords:** gait analysis; foot progression angle; APDM; inertial measurement unit; toe-in; toe-out

## 1. Introduction

Biomechanical interventions can be used to assist in the management of chronic conditions, and interest has emerged around gait retraining interventions for people with medial knee osteoarthritis (KOA) [1]. The most researched intervention with a demonstrated ability to reduce indices of knee load is walking with an altered foot progression angle (FPA). FPA is defined as the angle of the longitudinal axis of the foot relative to the line of the body's over-ground progression during gait [2]. Changing the FPA to either an internally oriented position (toe-in) or an externally oriented position (toe-out) has been shown to reduce pain in KOA patients after a six-week gait retraining program [2,3].

For gait retraining interventions to be clinically implemented, valid and reliable measurement tools need to be available in a clinical setting. Currently, FPA is measured in a variety of ways. The most comprehensive method is 3D motion capture using markers on the foot tracked by infrared cameras [4]. This is commonly considered as the gold standard for the tracking of human movement [5]. However, these motion capture systems are costly, require a fixed laboratory, are sensitive to optical occlusion, and require a time-consuming analysis to yield results. Alternatively pressure sensitive walkways such as the GAITRite® (Franklin, TN, USA), the Strideway™ (Tekscan, Boston,

MA, USA), and the Zeno™ Walkway (ProtoKinetics, Havertown, PA, USA) can collect and analyze spatiotemporal parameters from foot pressure data, including the FPA [6]. Wearable inertial measurement units (IMU) include an accelerometer, a magnetometer, and a gyroscope. A wide range of IMUs, each featuring their own distinctive characteristics, are commercially available from companies such as APDM (Portland, OR, USA), Xsens Technologies B.V. (Enschede, The Netherlands), Technaid S.L. (Madrid, Spain), IMeasureU (Auckland, New Zealand), and Noraxon (Scottsdale, AZ, USA). IMUs are insensitive to occlusion and are not limited to a specified lab or examination room. The challenges of using IMUs may include sensitivity to magnetic disturbances, power use (the sensors are wireless), and their dependence on complex proprietary sensor fusion algorithms and correction software. When placed on the lower extremities, IMUs can measure spatiotemporal data, including the FPA [7]. Therefore, IMUs seem particularly well suited for measuring the FPA in a wide range of clinical practices that aim to apply gait retraining in patients with KOA.

Although motion capture is considered the gold standard, a pressure sensitive walkway (PSW) designed to measure foot imprints is considered to perform equally well [8]. Yielding valid FPA measurements using IMUs is dependent on both the proper attachment of the IMU to the foot (for anatomical calibration) and accurate signal processing of the sensor signals. To assess whether IMUs are a viable option for the attainment of FPA measurements, the validity and reliability of these measurements need to be established. The purpose of this study is, therefore, to determine the reliability and validity of FPA measurements based on IMUs, by comparing IMU performance in FPA analyses to the highly accurate and established PSW methodology, using the gait strategies that are used for gait retraining in KOA.

## 2. Materials and Methods

### 2.1. Study Design

This was a test-retest reliability and validity study, to evaluate FPA measured by IMUs against a PSW serving as the reference standard. This study was approved by the university's institutional review board, and all participants provided written informed consent prior to participation.

### 2.2. Participants

Twenty healthy adults (14 females, 6 males) were recruited from the university and surrounding community via electronic media and word of mouth in April 2019. Only participants who could walk independently without aids and complete at least 10 trials of walking for 90 s on a flat surface were included in the study. Participants were excluded if they had pain that affected lower limb movements, neurological conditions affecting gait or balance, or were unable to understand and speak English. To obtain an intraclass correlation (ICC) of 0.7 or higher, a sample size of at least 18 participants was required [9]. As this study only investigated the reliability and validity of the instruments and not participant-related response, each leg was considered in the analysis. Therefore, 40 legs from 20 participants were evaluated in this study.

### 2.3. Procedures

#### 2.3.1. Instrumentation

Concurrent gait data were collected using two systems: (i) the Zeno™ Walkway (ProtoKinetics, Havertown, PA, USA), a pressure sensitive walkway; and (ii) Opal sensors (APDM, Portland, OR, USA), a wearable wireless IMU-based system. The PSW had a width and length of 1.22 m and 3.66 m, respectively (4′ by 12′), and was wired to the host computer to communicate via ProtoKinetics Movement Analysis Software (version 507.C7c, PKMAS, ProtoKinetics, Havertown, PA, USA). For the IMU-based system, 7 sensors were used: 1 lumbar, 2 upper leg, 2 lower leg, and 2 feet sensors. A more detailed description of the locations can be found in Table 1. The system makes

use of a docking station to charge and configure the sensors and a wireless access point to communicate between the sensors and the host computer. The sensors are shaped similarly to a watch, of a small size (55 mm × 40.2 mm × 12.5 mm), and weigh 25 g. The sensors recorded movement with triaxial accelerometers, gyroscopes, and magnetometers [10]. The PSW measured at a sampling rate of 120 Hz and the IMUs at 128 Hz using the Moveo Explorer software, (version 1.0.0.201904110002, APDM, Portland, OH, USA). All data were collected and combined in Matlab R2018b (MathWorks®, Natick, MA, USA) and for statistical calculations IBM SPSS Statistics for Windows, version 25 (IBM Corp., Armonk, NY, USA) was used.

**Table 1.** Number of sensors, positioning, and description of position of the IMUs (Opal sensors) used.

| Number of Sensors | Positioning | Description of Position |
| :---: | :---: | :--- |
| 1 | Lumbar | Centered on the low back, at the base of the spine. Superior aspect of the posterior sacral surface. |
| 2 | Upper leg | Lateral aspect of thigh, midline right over the iliotibial band, between the muscular tissue, one hand's width above the knee. |
| 2 | Lower leg | Medial to the front of the tibia, on the flat surface of the bone, high enough for the strap to wrap just above the widest part of the calf muscle. |
| 2 | Foot | Centered on top of the foot, aligned with the second metatarsal. |

### 2.3.2. Assessment

After the sensors were attached to the participants as described in Table 1, participants were allowed to get comfortable perambulating on the walkway. All participants were asked to wear their own comfortable shoes. The data collection protocol consisted of four walking trials of 90 s each. During the first two trials, participants were asked to walk using their "natural gait" (normal walking). Between the trials, the IMUs were taken off and attached again, for test-retest measurements. For the next trial, participants were asked to walk with a toe-out gait, which consisted of walking with the toes pointing outwards as far as comfortably possible. For the last trial, participants were asked to do the opposite and point their toes inwards (toe-in gait). Each trial started with the participant standing still in the calibration pose (with the longitudinal axis of the feet perpendicular to the coronal plane), to ensure IMU calibration, after which both measurement systems were started simultaneously. Participants walked for 90 s back and forth on a 10-m-long path, including the 3.66 m walkway. After 90 s, both systems were stopped concurrently.

### 2.3.3. Outcome Measures

Moveo Explorer (IMUs) measures the FPA as *"the lateral angle of the foot during the stance phase, relative to the forward motion of the foot during the swing phase"* [11]. In PKMAS software (walkway), the algorithm creates an ellipse with the smallest area that first completely encloses all the activated sensors of a footprint. Then, the direction of the foot is defined by the long axis of the ellipse. The FPA by PKMAS is measured as *"the angle between the Direction Of Progression (DOP) and the Foot Angle (degrees)"* [12]. Positive FPA values indicate toes pointing outwards, whereas negative values indicate the toes pointing inwards. Values below $-40°$ or above $40°$ were assumed to be errors and were deleted from the dataset.

### 2.4. Statistical Analysis

Reliability was assessed using intraclass correlation coefficient (ICC) and standard error of measurement (SEM). ICC and SEM were calculated between the first two trials of the protocol (both normal walking), to assess the test-retest reliability for each measurement system. For further calculations, the second trial was used as a baseline, because the IMUs

were kept in place from the second trial onwards. ICC estimates and their 95% confidence intervals were calculated based on a mean-rating (k = 2), absolute-agreement, 2-way mixed-effects model. The SEM was calculated as $\text{SEM} = \text{SD} * \sqrt{1 - R_{xx}}$, using the standard deviation (SD) and the reliability of the test ($R_{xx}$). The SEM value can range from 0 to the value of the standard deviation, with a higher value indicating a lower test reliability. The ΔFPA is calculated by subtracting the baseline FPA per leg from the gait strategy (toe-out gait or toe-in gait). Finally, the minimal detectable change (MDC) was determined by $\text{MDC} = 1.96 * \text{SEM} * \sqrt{2}$. The MDC is described as the least amount of change that is not the result of measurement error [13].

Validity was assessed by calculating repeated measures ANOVA (analysis of variance) test, ICC, and Bland–Altman plots. ANOVA tests were performed with one dependent variable, FPA, and two independent variables: "gait type" (baseline/toe-in/toe-out) and "measurement instrument" (IMU/PSW). Mauchly's test indicated that the assumption of sphericity had been violated for both independent variables. Greenhouse–Geisser corrections were applied to the degrees of freedom, such that a valid critical F-value could be obtained. ICC was calculated per system for the two baseline measurements. Bland–Altman plots were created using the mean and the difference in FPA between both measurement systems. Then, 95% confidence intervals (95% CI) were added as the limits of agreement. Finally, a scatterplot was created using the mean and the difference in FPA between the systems for all gait types, combined with a linear regression to identify proportional bias.

## 3. Results

Overall, participants had a mean age of 33.7 years (SD = 10.3 years) and were of normal weight [14] on average (body mass index (BMI) = 23.4 kg/m$^2$, SD = 3.8).

### 3.1. Outcome Measures

Mean FPA measurements were highest in the toe-out condition and most negative during toe-in gait. On average IMUs responded with a magnitude of around 15 degrees to both conditions, while the PSW measured nearly 3 degrees less (Table 2). In total, 7974 steps were detected. While, 63 steps were removed from the dataset, as they fell outside the acceptable range of values (between −40° and 40°); i.e., almost physically impossible in healthy people; 86% of these removed steps were measured by the PSW.

**Table 2.** Mean foot progression angle (FPA), standard deviation (SD), 95% confidence intervals (CI), minimum and maximum values and mean difference in FPA between the two systems in three conditions: baseline gait, toe-out gait, and toe-in gait. All values are in degrees for IMUs (Opal sensors) and the PSW (Zeno™ walkway). The calculations are based on 7974 steps by 20 participants.

| Instrument | Variable | Baseline | Toe-Out Gait | Toe-In Gait |
|---|---|---|---|---|
| IMUs | Mean FPA ± SD | 5.6 ± 4.9 | 19.9 ± 6.3 | −9.9 ± 6.8 |
| | (95% CI) | (4.0: 7.1) | (17.9: 21.9) | (−12.1: −7.7) |
| | Min; Max | −2.0; 19.5 | 8.6; 31.5 | −29.1; 5.5 |
| PSW | Mean FPA ± SD | 5.2 ± 5.5 | 17.3 ± 6.0 | −7.2 ± 5.4 |
| | (95% CI) | (3.5: 7.0) | (15.4: 19.2) | (−9.0: −5.5) |
| | Min; Max | −4.5; 21.4 | 9.1; 33.5 | −18.4; 6.4 |
| IMU-PSW | Mean ΔFPA between systems ± SD | 0.3 ± 3.6 | 2.6 ± 3.7 | −2.7 ± 4.1 |
| | (95% CI) | (−0.82: 1.5) | (1.4: 3.8) | (−4.0: −1.3) |
| | Min; max | −11.3; 7.3 | −4.4; 11.7 | −13.0; 5.2 |
| | *p*-value | 0.57 | *p* < 0.001 | *p* < 0.001 |

### 3.2. Reliability

The ICCs for the walkway indicate excellent reliability, and the ICCs for IMUs indicate good reliability (Table 3) [15]. The error for the IMUs (SEM = 1.6°) was larger compared to the error of the PSW (SEM = 0.96°), which resulted in a higher MDC for the IMUs. In Table 4, the ΔFPA between the different gait types is evaluated for both measurement systems. The IMUs detected a larger gait alteration between baseline and the gait modification strategies than the PSW. For both systems, the gait alteration between baseline and toe-out gait was significantly positive and significantly negative between the baseline and toe-in gait.

**Table 3.** Assessment of baseline test-retest intraclass correlation coefficients (ICC) with 95% confidence intervals, standard deviation (SD), standard error of measurement (SEM), and minimal detectable change (MDC) in degrees.

| | IMUs Baseline Test-Retest | PSW Baseline Test-Retest |
|---|---|---|
| ICC absolute agreement | 0.89 | 0.97 |
| (95% CI) | (0.79; 0.94) | (0.95; 0.99) |
| *p*-value | $p < 0.001$ | $p < 0.001$ |
| SEM (°) | 1.6 | 0.96 |
| MDC (°) | 4.5 | 2.7 |

**Table 4.** ΔFPA in degrees for toe-out gait and toe-in gait for both systems, and difference in ΔFPA between the systems. ΔFPA is calculated by subtracting the baseline FPA from the toe-out gait or toe-in gait FPA. All variables are in degrees.

| | IMUs | | PSW | | IMUs—PSW | |
|---|---|---|---|---|---|---|
| Variable | ΔFPA toe-out | ΔFPA toe-in | ΔFPA toe-out | ΔFPA toe-in | Difference in ΔFPA toe-out | Difference in ΔFPA toe-in |
| Mean FPA ± SD | 14.3 ± 5.4 | −15.5 ± 6.7 | 12.0 ± 4.8 | −12.5 ± 5.1 | 2.3 ± 2.4 | −3.0 ± 3.6 |
| Median | 14.4 | −14.1 | 11.5 | −11.6 | 2.3 | −2.5 |
| Min; max | 4.0; 26.6 | −41.6; −7.0 | 2.3; 25.1 | −30.6; −4.1 | −3.1; 7.3 | −11.7; 2.9 |
| 95% CI | 12.6; 16.0 | −17.6; −13.3 | 10.5; 13.6 | −14.1; −10.9 | 1.5; 3.0 | −4.2; −1.8 |

### 3.3. Validity

The repeated measures ANOVA resulted in a significant main effect of the variable "gait type" ($p < 0.001$). The results show there was no significant effect of the variable "measurement instrument" ($p = 0.875$). The ICC for correlation between the two systems for baseline measurements was indicative of good correlation, being 0.87 and 0.84 when outliers were included. Bland–Altman plots are shown in Figure 1 for all gait types, with mean differences between the two systems and limits of agreement. Data were checked for heteroscedasticity, but Kendall's tau (τ) was negative, so the data were considered homoscedastic [16], meaning the observed variance is independent of the variable mean [17]. A scatterplot is shown in Figure 2 of the mean FPA against the difference in FPA between the two systems of all gait types combined. A significant ($p < 0.001$) regression line was fitted to the dots with the following equation: $y = -0.79 + 0.17 * x \left( R^2 = 0.234 \right)$.

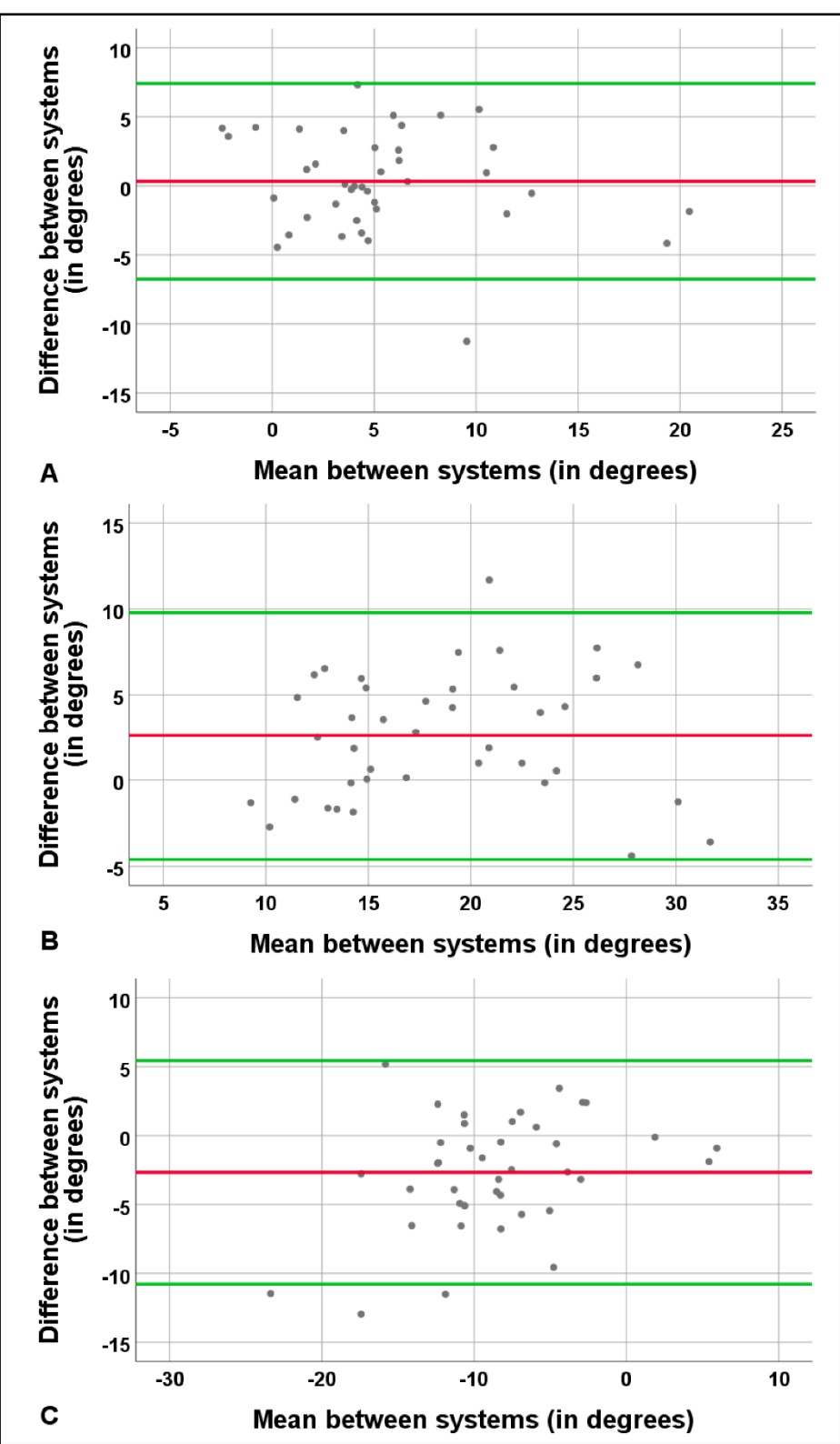

**Figure 1.** Bland–Altman plots for baseline (**A**), toe–out (**B**), and toe–in (**C**) gait. Every grey dot represents one leg. Difference in FPA between IMUs (Opal sensors) and the PSW (Zeno™ Walkway) is displayed on the y–axis, and the mean FPA for Opal and Zeno™ is displayed on the x–axis. The red line represents the mean difference and the green lines the 95% limits of agreement.

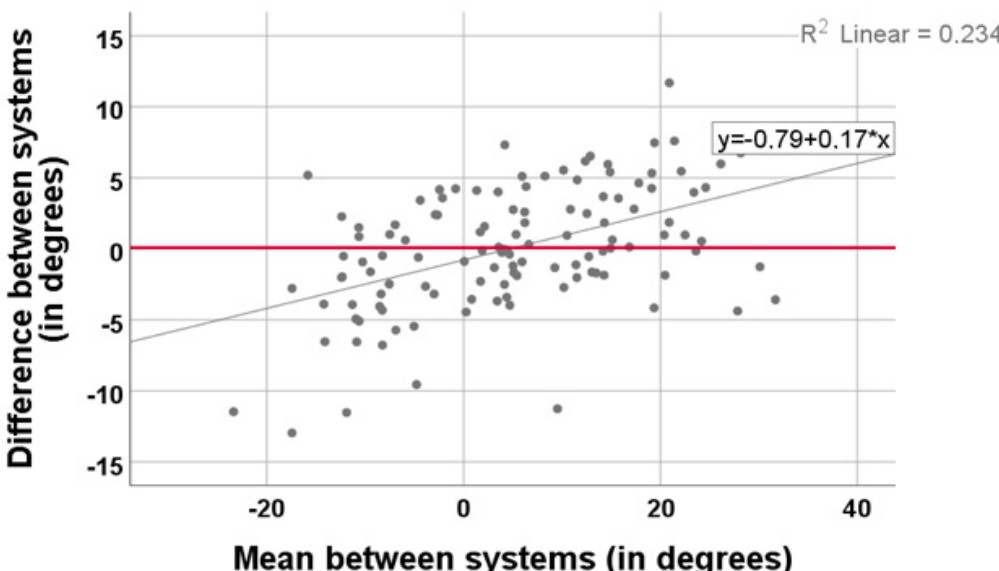

**Figure 2.** Scatterplot of mean FPA against the difference in FPA between the two systems of all gait types combined. Difference in FPA between IMUs (Opal sensors) and the PSW (Zeno™ Walkway) is displayed on the y–axis, and the mean FPA for Opal and Zeno™ is displayed on the x–axis. The red line represents the mean difference. A linear regression line ($y = -0.79 + 0.17 * x$) is fitted to the dots.

## 4. Discussion

### 4.1. Summary of the Main Findings

This study included a reliability and validity study of the Opal sensors (IMUs) against the Zeno™ Walkway (PSW). The current data demonstrates that both measurement systems are reliable for measuring FPA with natural, toe-in, and toe-out gaits. Moreover, the systems have a good agreement in baseline conditions, but a significant difference of 2.7 degrees between the systems was found when performing the gait modification strategies. Nevertheless, these findings are promising for clinicians, to have confidence in the use of IMUs for gait retraining.

### 4.2. Outcome Measures

As expected, the mean FPA was most positive for toe-out gait compared to baseline and toe-in gait. Toe-in gait yielded the most negative angle. In this study, the target angle was a FPA that felt unnatural but not uncomfortable. A mean FPA deviation of approximately $12°$ to $16°$ was achieved in this way. In a previous study the FPA target angle was a deviation of $10°$ from the baseline FPA [18]. The FPA data from this study agree with the FPA in healthy participants in previous research [6].

### 4.3. Reliability

Both systems seem to be reliable measurement systems for quantifying FPA. The baseline test-retest reliability for the PSW was excellent (ICC = 0.97), and for the IMUs it was considered to have good reliability (ICC = 0.89). When outliers were included, the PSW test-retest ICC equaled 0.90, which indicates good reliability; the inclusion of outliers did not affect the test-retest ICC for IMUs. From this data, it can be concluded that both systems have a high reproducibility, when walking with a natural FPA. Previous research showed a similar test-retest ICC of 0.98 [19] for FPA during normal walking on the GAITRite®, a pressure sensitive walkway similar to the Zeno™ walkway that was used in this study. Another study found a lower test-retest reliability for FPA measured with the GAITRite® in older adults (aged between 76 and 87 years), ICC = 0.71 (right foot) and ICC = 0.82 (left foot), compared to young adults (aged between 22 and 40 years) ICC = 0.88 (right foot) and

ICC = 0.94 (left foot) [8]. This may be the result of higher gait variability in older adults and should be considered when using these systems for other populations.

After excluding outliers, the SEM for the PSW is smaller (0.96°) than for the IMUs (1.62°), but the latter can still be considered acceptable, particularly in a clinical environment. The SEM is the amount of error that can be considered as measurement error. The MDC indicated that, consequently, IMUs are not suitable for precise measurements smaller than 5°, with the PSW not being suitable for measurements smaller than 3°. The MDC is described as the least amount of change that is not the result of measurement error [13]. However, it should be noted that not every step is identical, so part of this MDC is explained by physiological differences. A t-test showed a significant difference between the two measurement systems for toe-in -and toe-out gait.

### 4.4. Validity

Both the IMUs and PSW detected changes in FPA with each gait modification strategy implemented. The mean difference between the two measurement systems in baseline conditions was negligible (0.33°). IMUs seemed to amplify the ΔFPA compared to the PSW for the gait modification strategy conditions by 15–30%. A significant difference of 2.7 degrees between the systems was found when performing the gait modification strategies. A similar study found a comparable difference in FPA of 2.6 degrees between a foot-worn inertial sensor and a motion capture system [20]. It is unclear whether the location of the IMU in detecting deformation of the foot (when loading in extreme positions) could have caused a varus or valgus effect on the FPA. ΔFPA amplification could also have been caused by sensor movement, since the lumbar and foot sensors were placed above clothing and shoes. Sensors were secured as firmly as possible, but some unwanted movement may have been possible. When using the systems interchangeably, which is not recommended, one should apply a correction for FPA for toe-out and toe-in gait.

The Bland–Altman plot for baseline gait indicates a good agreement between the two measurement systems. The other two Bland–Altman plots (toe-out gait and toe-in gait) show a proportional bias. The IMUs amplified the values (more negative with toe-in gait and more positive with toe-out gait) compared to the PSW values. This can be seen in the scatterplot with all gait types combined in Figure 2. The regression line shows a positive slope and predicts the difference between the systems significantly well ($B = 17$, $p < 0.001$). The $R^2$ value shows that 23% of the total variation in the difference between the systems can be explained by the mean between the systems. The limits of agreement of the Bland–Altman plots stayed proportionate, the variability consistent, and with negligible outliers, suggesting good validity during implementation of gait modification strategies.

A similar study, comparing spatiotemporal gait parameters between the Opal sensors and the GAITRite®, found a comparable trend, where the Opal sensors consistently overestimated the FPA measurements compared to the GAITRite® values, which increased as the gait variability increased [21]. Another study, comparing another wearable IMU system to 3D motion capture found an ICC of 0.94 between the two systems, which is higher than the ICC found in this study (ICC = 0.87) [7]. However, the other study used target FPAs with visual feedback, which would reduce the amount of individual gait variation.

### 4.5. Strengths and Limitations

The strengths of this study included the adequate sample size and many steps, as a result of the relatively long measurements in multiple conditions. The use of test-retest reliability makes the evidence more credible. The protocol of testing validity under different types of gait is important because those are the conditions used in a gait retraining session, which makes the results clinically meaningful.

The findings of this study should, however, be interpreted in the context of two main limitations. First, all participants were healthy and were asked to walk with an increased toe-in or toe-out angle for the last two conditions. Thus, the accuracy may differ in people with a movement disorder, e.g., foot drop. Second, at the end of the 10-m-long path

participants had to turn around each time, until a duration of 90 s was achieved. These turns were only recorded by the IMUs and were automatically deleted from the dataset by the APDM software, but these turns could have influenced the participants' gait during the few steps between the PSW and the turn. As described before, APDM and PKMAS have a different definition and measurement of the FPA. The line of progression is defined by APDM when the foot is in swing phase, and by PKMAS it is calculated on the basis of footprints. Besides the differences in the measurement precision, the difference in definition may also have been a source of the differences in FPA.

*4.6. Recommendations for Future Studies*

The findings from this study provide support for accurate and reliable measurement of FPA during gait using Opal IMUs, and are specifically appropriate for gait retraining therapy clinical scenarios. Further research should be conducted with the acquisition of the FPA using dedicated IMU FPA algorithms or the use of artificial intelligence, to make more reliable estimates. Future research on IMUs should also focus on determining optimal FPA feedback methods for participants undergoing gait retraining. Some research has already been conducted with 3D motion analysis and a proof-of-concept a haptic feedback-sensorized shoe in combination with a target FPA [22]. Clinical studies of long-term duration are needed to determine the best methods to change gait in people with knee OA.

**5. Conclusions**

The results suggest that using the Opal sensors as IMUs are sufficiently reliable and valid to measure FPA in gait retraining. There were small systematic differences compared to the reference standard that should be accounted for by the interpretation. IMUs provide a promising tool for clinicians and researchers aiming to quantify FPA for gait retraining.

**Author Contributions:** All authors made substantial contributions to the following: (1) Conceptualization, F.C.A.U., J.G. and M.S. (2) methodology, F.C.A.U., J.G., J.H. and M.S. (3) formal analysis, F.C.A.U. (4) investigation, F.C.A.U. (5) writing—original draft preparation, F.C.A.U. (6) writing—review and editing, J.G., J.H. and M.S. (7) visualization, F.C.A.U. (8) supervision, J.G., J.H. and M.S. (9) project administration, M.S. All authors have read and agreed to the published version of the manuscript.

**Funding:** This research received no external funding.

**Institutional Review Board Statement:** The study was approved by the Human Research Ethics Committee of the UNIVERSITY OF SYDNEY (Project No 2018/969, date of approval 19 February 2019).

**Informed Consent Statement:** Informed consent was obtained from all subjects involved in the study.

**Data Availability Statement:** The data presented in this study are available on request from the corresponding author. The data are not publicly available, due to ethical limitations.

**Conflicts of Interest:** There are no financial or personal interests to disclose that could have potentially and inappropriately influenced the integrity of the work in this manuscript, by any of the authors.

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
