# Peer review of "Reliability and Validity of IMU-Based Foot Progression Angle Measurement under Different Gait Retraining Strategies"

_applsci, doi:10.3390/app12136519_

Round 1

Reviewer 1 Report

The study evaluates the reliability and validity of an IMU
system to measure FPA under different gait retraining strategies. IMUs are insensitive to occlusion and are not limited to a specified lab or examination room, in comparison with 3D motion capture using markers
on the foot tracked by infrared cameras, and IMUs are less expensive in comparison with 3D motion systems. Twenty healthy adults completed at least 10 trials of walking for 90 seconds on a flat surface were included in the study. Overall, participants had a mean age of 33.7 years (SD=10.3 years) and were of normal weight. The study included a reliability and validity study of the Opal sensors (IMUs) against the ZenoTM
Walkway (PSW). The results suggest that using the Opal sensors as IMUs are sufficiently reliable and valid to measure FPA in gait retraining.  IMUs provide a promising tool for clinicians and researchers aiming to quantify FPA for gait retraining .

There are minor format details that must be addresed:

A) Table 3 must be placed in the same page.

B) The resolution of Figures 1 and 2 must be improved. A large font size must be used on labels on both axis. In the third plot of figure 1, the x labels are not visible. I suggest to use a subfigure environment to remove the "header" of each plot of figure 1 (and in figure 2 too).

C) Some references are old and they should be updated. The following references should be added to the manuscript:

Rose M, Costello K, Eigenbrot S, Torabian K, Kumar D
Inertial Measurement Units and Application for Remote Health Care in Hip and Knee Osteoarthritis: Narrative Review
JMIR Rehabil Assist Technol 2022;9(2):e33521
URL: https://rehab.jmir.org/2022/2/e33521
DOI: 10.2196/33521

T. Tan, Z. A. Strout, H. Xia, M. Orban and P. B. Shull, "Magnetometer-Free, IMU-Based Foot Progression Angle Estimation for Real-Life Walking Conditions," in IEEE Transactions on Neural Systems and Rehabilitation Engineering, vol. 29, pp. 282-289, 2021, doi: 10.1109/TNSRE.2020.3047402.

Joe Verghese, Roee Holtzer, Richard B. Lipton, Cuiling Wang, Quantitative Gait Markers and Incident Fall Risk in Older Adults, The Journals of Gerontology: Series A, Volume 64A, Issue 8, August 2009, Pages 896–901, https://doi.org/10.1093/gerona/glp033

Hopkins, W.G. Measures of Reliability in Sports Medicine and Science. Sports Med 30, 1–15 (2000). https://doi.org/10.2165/00007256-200030010-00001

Atkinson, G., Nevill, A.M. Statistical Methods For Assessing Measurement Error (Reliability) in Variables Relevant to Sports Medicine. Sports Med 26, 217–238 (1998). https://doi.org/10.2165/00007256-199826040-00002

Author Response

Thank you for your comments and the opportunity to address them. Please find the responses in red.

Comments and Suggestions for Authors

The study evaluates the reliability and validity of an IMU system to measure FPA under different gait retraining strategies. IMUs are insensitive to occlusion and are not limited to a specified lab or examination room, in comparison with 3D motion capture using markers on the foot tracked by infrared cameras, and IMUs are less expensive in comparison with 3D motion systems. Twenty healthy adults completed at least 10 trials of walking for 90 seconds on a flat surface were included in the study. Overall, participants had a mean age of 33.7 years (SD=10.3 years) and were of normal weight. The study included a reliability and validity study of the Opal sensors (IMUs) against the ZenoTM Walkway (PSW). The results suggest that using the Opal sensors as IMUs are sufficiently reliable and valid to measure FPA in gait retraining.  IMUs provide a promising tool for clinicians and researchers aiming to quantify FPA for gait retraining .

There are minor format details that must be addressed:

Table 3 must be placed in the same page

 Response: Table 3 is now on the same page

The resolution of Figures 1 and 2 must be improved. A large font size must be used on labels on both axis. In the third plot of figure 1, the x labels are not visible. I suggest to use a subfigure environment to remove the "header" of each plot of figure 1 (and in figure 2 too).

Response: The resolution of Figures 1 and 2 have been improved as suggested.

Some references are old and they should be updated. The following references should be added to the manuscript:

Rose M, Costello K, Eigenbrot S, Torabian K, Kumar D Inertial Measurement Units and Application for Remote Health Care in Hip and Knee Osteoarthritis: Narrative Review
JMIR Rehabil Assist Technol 2022;9(2):e33521
URL: https://rehab.jmir.org/2022/2/e33521 DOI: 10.2196/33521

Tan, Z. A. Strout, H. Xia, M. Orban and P. B. Shull, "Magnetometer-Free, IMU-Based Foot Progression Angle Estimation for Real-Life Walking Conditions," in IEEE Transactions on Neural Systems and Rehabilitation Engineering, vol. 29, pp. 282-289, 2021, doi: 10.1109/TNSRE.2020.3047402.

Joe Verghese, Roee Holtzer, Richard B. Lipton, Cuiling Wang, Quantitative Gait Markers and Incident Fall Risk in Older Adults, The Journals of Gerontology: Series A, Volume 64A, Issue 8, August 2009, Pages 896–901, https://doi.org/10.1093/gerona/glp033

Hopkins, W.G. Measures of Reliability in Sports Medicine and Science. Sports Med 30, 1–15 (2000). https://doi.org/10.2165/00007256-200030010-00001

Atkinson, G., Nevill, A.M. Statistical Methods For Assessing Measurement Error (Reliability) in Variables Relevant to Sports Medicine. Sports Med 26, 217–238 (1998). https://doi.org/10.2165/00007256-199826040-00002

Reviewer 2 Report

General Comments

This is an interesting and very well executed study comparing two different methods of measuring foot progression angle (FPA). Therapeutic adjustment the FPA during gait may result in a reduction of pain in patients suffering from OA of the knee. The methods compared for measuring FPA were a standard of care pedobarographic system (PSW) and a 9DOF inertial measurement unit (IMU) which is strictly speaking an AHRS system using sensor fusion). IMU technology has advanced in recent years, and the promise of such systems are that they will provide off-site kinematic measurements without the need for expensive additional equipment, such as a gait-lab or a PSW. Although IMU hold much promise, they are very complex devices that may experience drift of the measured angles and are sensitive to magnetic field inhomogeneity, which is present in almost all buildings. Scientific investigations of measurements based on IMU in comparison with standard methods are thus of particular importance and relevance. Healthy subjects were observed using both measurement methods during normal gait (baseline), as well as when walking during a toe-in and toe-out foot posture. Reliability of both methods, as well as the validity of the IMU measurements in comparison to PSW were investigated. The authors concluded that the IMU were a promising alternative for clinical application with some limitations. Although the study is of interest and well executed, some issues in particular with regards to the presentation and interpretation of the data remain.

Specific Comments

L50: The comparison of standard 3D motion capture and IMU systems would profit from a bit more objectivity, and should include some of the limitations/challenges associated with the use of IMU (magnetic field variations, complex proprietary sensor fusion algorithms and correction software). This is perhaps what the authors state later when they refer to “accurate signal processing” without further elucidation.

L154: Describe the results, not the table, i.e. refer to Table parenthetically.

L159 (Table 2): The table heading is a run-on sentence and difficult to read. Analysis seems to pool left and right legs; this should be mentioned in the methods. All calculations based on “7974 steps by 40” legs is not particularly informative, and should be reduced to per subject and condition (N of the respective condition).

L168: is it error, or rather variability. The SEM is directly related to the group size (N), hence probably not a good measure.

Table 4: what is mean of IMU and PSW telling us here (other than to be used in the B-A diagrams)?

L189: Bland-Altman are shown in Figure 1: yes, but what do the B-A diagrams tell us. Describe the results of the B-A not the figure, that’s what the figure headings are for. No need to repeat the figure-headings in the text. Are you seeing bias in the B-A diagram, are the differences observed (clinically) relevant?

L193: ditto. Ok, nice, you generated a scatterplot and fitted a regression line. It is your task to interpret it, not the readers. Please don’t rely on the reader to interpret the data, interpret the data and leave it to the reader to agree or disagree with your conclusions.

L213: in my opinion the B-A diagrams indicate relatively large CI and suggest that the methods are probably not be clinically equivalent. This may also be due to the different kinematic definitions of the method (relative to “forward motion in the swing phase of the foot” for the IMU and “direction of progression”. The variability of the swing phase from patient to patient may be confounding the IMU-based measure of FPA.

Figure 1: the borders of the diagrams are cut off, grid lines show through text.

L226: removing outliers from data must be justified by some unexpected or unwanted event, for instance a problem with the method (device), or the patient/subject (stumbling etc.). The removal of outliers solely based on the definitions of outliers in a box-plot is not justified because recursive removal of outliers will introduce bias into the data.

L234: increased variability of patient gait is not necessarily related to decreased reliability of a measurement method.

L236: see above with regards to outliers.

4.3 Validity: while the mean difference is quite small (0.33 deg), the B-A acutally indicate quite large ranges of the 95% limits of agreement, from about -11 to about 6 deg for toe-in gait. This is IMHO illustrative and a valid and justified result, but that range of a difference in between the two methods might be clinically relevant in many contexts.

L285: you mention the possible effect of turning on IMU data, perhaps you could have reduced the IMU data temporally to the middle of the interval between turns to rule out this effect and also to compare data of the patient at about the same time (i.e. when passing over the PSW).

Author Response

Reviewer 2Thank you for your comments and the opportunity to address them. Please find the responses in red.

General Comments

This is an interesting and very well executed study comparing two different methods of measuring foot progression angle (FPA). Therapeutic adjustment the FPA during gait may result in a reduction of pain in patients suffering from OA of the knee. The methods compared for measuring FPA were a standard of care pedobarographic system (PSW) and a 9DOF inertial measurement unit (IMU) which is strictly speaking an AHRS system using sensor fusion). IMU technology has advanced in recent years, and the promise of such systems are that they will provide off-site kinematic measurements without the need for expensive additional equipment, such as a gait-lab or a PSW. Although IMU hold much promise, they are very complex devices that may experience drift of the measured angles and are sensitive to magnetic field inhomogeneity, which is present in almost all buildings. Scientific investigations of measurements based on IMU in comparison with standard methods are thus of particular importance and relevance. Healthy subjects were observed using both measurement methods during normal gait (baseline), as well as when walking during a toe-in and toe-out foot posture. Reliability of both methods, as well as the validity of the IMU measurements in comparison to PSW were investigated. The authors concluded that the IMU were a promising alternative for clinical application with some limitations. Although the study is of interest and well executed, some issues in particular with regards to the presentation and interpretation of the data remain.

Specific Comments

L50: The comparison of standard 3D motion capture and IMU systems would profit from a bit more objectivity, and should include some of the limitations/challenges associated with the use of IMU (magnetic field variations, complex proprietary sensor fusion algorithms and correction software). This is perhaps what the authors state later when they refer to “accurate signal processing” without further elucidation.

Response: The following information has been updated on L49 - 51 - Challenges of using the IMUs could include sensitivity to magnetic disturbances, power use (the sensors are wireless) and its dependency complex proprietary sensor fusion algorithms and correction software

L154: Describe the results, not the table, i.e. refer to Table parenthetically.

Response: We have updated the description of results for the outcome measures to include the following information (starting L153) -

Mean FPA measurements were highest in the toe-out condition and most negative during toe-in gait. On average IMUs responded with a magnitude of around 15 degrees to both conditions, while the PSW measured nearly 3 degrees less (Table 2).

L159 (Table 2): The table heading is a run-on sentence and difficult to read. Analysis seems to pool left and right legs; this should be mentioned in the methods. All calculations based on “7974 steps by 40” legs is not particularly informative, and should be reduced to per subject and condition (N of the respective condition).

Response: The Table 2 heading (L160) has been updated to the following information –

Mean foot progression angle (FPA), standard deviation (SD), 95% confidence intervals (CI), minimum and maximum values and mean difference in FPA between the two systems in three conditions: baseline gait, toe-out gait and toe-in gait. All values are in degrees for IMUs (Opal sensors) and the PSW (Zeno™ walkway). The calculations are based on 7974 steps by 20 participants.

L168: is it error, or rather variability. The SEM is directly related to the group size (N), hence probably not a good measure.

Response: Thank you for your comment. However, I think you are referring to the standard error of the mean, instead of the standard error of the measurement which is not directly related to N.

Table 4: what is mean of IMU and PSW telling us here (other than to be used in the B-A diagrams)?

Response: Thank you for your comment. The Mean IMU and PSW are indeed used to plot the B-A diagrams. We have now removed this information from Table 4 (and Table 2).

L189: Bland-Altman are shown in Figure 1: yes, but what do the B-A diagrams tell us. Describe the results of the B-A not the figure, that’s what the figure headings are for. No need to repeat the figure-headings in the text. Are you seeing bias in the B-A diagram, are the differences observed (clinically) relevant?

Response: Please see response to remark H

L193: ditto. Ok, nice, you generated a scatterplot and fitted a regression line. It is your task to interpret it, not the readers. Please don’t rely on the reader to interpret the data, interpret the data and leave it to the reader to agree or disagree with your conclusions.

Response: the regression clearly shows that the increased magnitude of the response to toeing-in and toeing-out of the IMU relative to PSW, is quite linear (in the presence of noise) Such predictability makes the systems not equivalent, but equally acceptable for clinical use.  

L213: in my opinion the B-A diagrams indicate relatively large CI and suggest that the methods are probably not be clinically equivalent. This may also be due to the different kinematic definitions of the method (relative to “forward motion in the swing phase of the foot” for the IMU and “direction of progression”. The variability of the swing phase from patient to patient may be confounding the IMU-based measure of FPA.

Response: We agree that the methods are not fully equivalent in all cases. However, the clinical meaningfulness, i.e to measure an toeing in or toe out, is still warranted in almost all cases, even with the relatively large CI (see also remark M).   

Figure 1: the borders of the diagrams are cut off, grid lines show through text.

Response: Thank you for pointing this out. This has now been fixed.

L226: removing outliers from data must be justified by some unexpected or unwanted event, for instance a problem with the method (device), or the patient/subject (stumbling etc.). The removal of outliers solely based on the definitions of outliers in a box-plot is not justified because recursive removal of outliers will introduce bias into the data.

Response: We completely agree with the reviewer that the reason for removing outliers should be an independent one. The argument for removing outliers (exceeding the -40° and 40° range) is that it is physically almost impossible in healthy people (added in line 162).

L234: increased variability of patient gait is not necessarily related to decreased reliability of a measurement method.

Response: We agree with the reviewer. People who have a dropfoot, for example, will influence the outcome of the zeno or the opals. Therefore, populations with additional impairments could influence the reliability of the system. Moreover, any increased variability is either measurement noise AND/OR (patho)physiological variability.

L236: see above with regards to outliers.

Response: see above

4.3 Validity: while the mean difference is quite small (0.33 deg), the B-A acutally indicate quite large ranges of the 95% limits of agreement, from about -11 to about 6 deg for toe-in gait. This is IMHO illustrative and a valid and justified result, but that range of a difference in between the two methods might be clinically relevant in many contexts.

Response: That is true but given the average response magnitude to toe-in (and toe-out) i.e 15 degrees, the sensor would still give meaningful feedback in many cases. We have updated our discussion (TODO) accordingly.

L285: you mention the possible effect of turning on IMU data, perhaps you could have reduced the IMU data temporally to the middle of the interval between turns to rule out this effect and also to compare data of the patient at about the same time (i.e. when passing over the PSW).

Response: The software of the Opals already deleted the turns out of the dataset. It was very hard to use the timing of the steps to compare the data because the Opals data doesn't give a timestamp of the steps.